# A Ban on Smartphone Usage during Recess Increased Children’s Physical Activity

**DOI:** 10.3390/ijerph18041907

**Published:** 2021-02-16

**Authors:** Charlotte Skau Pawlowski, Jonas Vestergaard Nielsen, Tanja Schmidt

**Affiliations:** Department of Sports Science and Clinical Biomechanics, University of Southern Denmark, Campusvej 55, 5230 Odense M, Denmark; Jvestergaard@health.sdu.dk (J.V.N.); tbschmidt@health.sdu.dk (T.S.)

**Keywords:** mobile phone, intervention study, physical activity, policy, child health, SOPLAY, questionnaire

## Abstract

School recess provides a unique opportunity for children to be active. However, many children perceive smartphones as a key barrier for engaging in physical activity during recess. The aim was to investigate if a ban on smartphone usage during recess changed children’s physical activity. During August–October 2020, children from grades 4–7 (10–14 years) at six Danish schools were banned from using their smartphones during recess for a four-week period. Questionnaire and systematic observation (SOPLAY) data were collected from 814 children before intervention (baseline) and 828 during the last week of intervention (follow-up). The mean frequency of physical activity significantly increased from baseline to follow-up (odds ratio = 1.370), as did physical activity on a moderate level (odds ratio = 1.387). Vigorous physical activity significantly decreased (odds ratio = 0.851). The increase in physical activity was found among both schools having outdoor and indoor recess, among both boys and girls, and nearly equally among grades 4–7. This suggests that implementing a ban on smartphone usage during recess would improve the everyday conditions for health among a broad range of schoolchildren. Future studies are needed to further investigate the association between recess physical activity and smartphone usage.

## 1. Introduction

School-aged children are known to have an increased use of smartphones that has become an integral part of their everyday lives [1,2]. An international study showed that Danish children are among the youngest (11.1 years) with a smartphone and the most frequent users of smartphones [3]. The last decades’ increase in smartphone usage has been blamed as a high-risk factor to the health of children [1,2]. In addition, a high number of schoolchildren internationally do not reach the recommended minimum level of 60 min of moderate-to-vigorous physical activity (PA) per day [4]. In Denmark, 74% of 11–15-year-old children do not reach the PA recommendations. Within this age group, girls are less active than boys and PA decreases significantly with age [5]. The health benefits of sufficient childhood PA include higher bone and muscle density, and lower blood pressure and risk of overweight and type 2 diabetes [6,7]. PA also has a positive effect on social health in schoolchildren [8]. It is important to help the children, who do not reach PA recommendations, to establish an active and healthy lifestyle in this phase of life. Being active during these years has been found to extend into adolescence and adulthood with long-term health implications [9,10].

School recess provides a unique opportunity for children to be active [11,12] and recess PA has been shown to significantly contribute to children’s overall level of PA [13]. Unfortunately, Danish children, participating in a large qualitative study, experienced smartphone usage as a key barrier for being active during recess [14]. Another study conducted in Sweden found a reduction in PA among 11- to 12-year-old boys from 2000 to 2013 and concluded that changes in recess habits, such as smartphone usage, may have influenced the PA reduction [15]. Nevertheless, to our knowledge, no intervention studies, neither in Denmark nor other countries, have directly investigated if a ban on smartphone usage during recess would change the children’s PA. This knowledge has been asked for by researchers [15] and practitioners such as school boards, health professionals, and politicians to inform future policies and actions [16,17]. Therefore, the aim of this study was to investigate if a four-week ban on smartphone usage during recess changed children’s PA.

## 2. Materials and Methods

### 2.1. Study Design

The study was a quasi-experimental intervention study, following the same group of children over time. Two different methodological approaches were used: Systematic observations and self-reported questionnaires. The two methods were used with a complimentary purpose and were equally weighted in the planning of the data collection, analyses, and discussion of results.

### 2.2. School Context

In Denmark, school is mandatory for children aged 6–16 years. Schools are typically organized into three tiers at the same location; junior (grades 0–3, 6–9 years old), middle (grades 4–6, 10–12 years old), and senior (grades 7–10, 13–16 years old). Approximately 60 min are dedicated to recess per school day, being distributed over a minimum of two main breaks: The morning break and the lunch break. Each school in Denmark has their own policies relating to recess. For example, it is up to the individual school to decide throughout the year if recess should occur outdoors or if the children can stay indoors during recess. For middle and senior students, recess is typically characterized by free play both indoors and outdoors without any organized curriculum or teacher involvement, except general schoolyard supervision. There is no national law on smartphone usage in schools and it is up to the individual school administration to decide whether smartphones should be banned or not.

### 2.3. Recruitment Procedure

In November 2019, we sent an invitation to all schools from 19 (out of 98) randomly selected municipalities in Denmark (145 schools). Nine schools wanted to participate, from which we selected six schools that met the inclusion criteria of not having implemented a ban on smartphone usage during recess. Five schools were public schools whereas one was a private school. The six schools varied in size (students enrolled), geographical location (region, urban/rural), and parental socioeconomic status (Table 1).

Children from grades 4–7 (10–14 years) were the target population of the study since PA decreases significantly during these years [18]. The schools decided for themselves which grades within the target population should be included in the study. See variation in the grades included in Table 1.

The study was conducted during the COVID-19 pandemic which caused changes in recess rules among the included schools. Most schools had introduced a policy obliging children to stay outdoors during all recesses, half of each recess, or at lunch break (Table 1). The schools had also made zones in the schoolyard and grade-divided the children into these zones to reduce socializing among different grades. The COVID-19 recess rules did not change between the two waves of data collection.

### 2.4. Intervention and Implementation Procedure

The purpose of the intervention was to implement a ban on screen usage (smartphones, tablets, and computers) during recess for the included grades. In the morning, all children placed their smartphones in a received “smartphone hotel”. The teachers were responsible for ensuring that all children handed in their smartphones, as well as for placing the “smartphone hotel” in a locked cupboard. Computers and tablets were told to be placed in the children’s school bags during recess. This study only includes data on smartphone usage since we registered almost no use of tablets and computers during recess across the six schools.

The intervention period at each school lasted for four weeks. The intervention was conducted in waves of two schools per week. Prior to the study start, an information process at each of the six schools took place to promote optimal implementation. At first, a meeting with the school principal was held to plan the intervention, followed by an information meeting with relevant school staff (e.g., grades 4–7 teachers). At the information meeting, the school staff were informed about the intervention procedures and measurements, and they had the possibility to ask questions about the study. They were also informed not to change any recess rules during the project period.

To ensure implementation, a procedure plan was handed out to each of the included teachers to help them to remember the specific procedure they needed to follow. Additionally, a reminder was e-mailed to the school principal a couple of days before the ban on screen usage had to be put into practice.

### 2.5. Data Collection

Data were collected at baseline, in the week prior to the intervention, and at follow-up in the last week of intervention (week 4). The entire data collection was conducted within seven weeks during August–October 2020.

The System for Observing Play and Leisure Activity in Youth (SOPLAY) was used for direct observations of the children’s PA and screen usage during recess. SOPLAY is a reliable and validated structured observation tool used to record movement patterns in a defined area such as a schoolyard [19,20]. The original version of SOPLAY was modified to fit the specific purpose of this study, by including information on screen usage (smartphone, tablet, computer). The data captured from SOPLAY were: PA intensity (sedentary, moderate, vigorous), smartphone usage (yes/no), gender, grade, weather (sunny, cloudy, windy, rainy), and temperature. To identify grade, the children wore a visible silicone bracelet (each grade with a different color). Prior to study start, the schools’ indoor and outdoor areas potentially used during recess by the participating children were identified and divided into 7–14 observation zones (depending on the size of the school area). We conducted four days of observations in total (two following days at baseline and the same two following days at follow-up), including two observation time points per day (morning break and lunch break). At the morning break each zone was observed once and at lunch break the zones were observed twice, due to longer breaks, resulting in 12 observations in each zone (six at baseline and six at follow-up). Each zone was observed for 5 min and started at the same time. The zones were observed by a team of research assistants, all trained in the same workshop to ensure alignment and reliable observations. Observations were recorded by hand in a scheme on a printed sheet.

A questionnaire was used to assess children’s self-reported PA and screen usage during recess. The data captured from the questionnaire were: Frequency of recess PA, frequency of smartphone usage during recess, name of school, grade, and gender. All questions were closed-ended questions, for example frequency Likert scale questions such as: “How frequently do you use a smartphone during recess on a weekly basis?” (4–5 days per week, 2–3 days per week, 1 day per week, never/rarely) and “How frequently do you do physical activity during the lunch break?” (4–5 days per week, 2–3 days per week, 1 day per week, never/rarely). The questionnaire was composed based on a selection of previous questionnaires, investigating children’s PA and screen media habits [21,22] and required approximately 15 min to complete. The questionnaire was pilot tested on the age group prior to the data collection, to ensure the questions were understandable and to test the length of the questionnaire. The questionnaire was electronically available since all Danish 4–7 graders have their own computer in school. To help children with difficulties with reading, they had the possibility to get the questions read out loud on their computer by clicking a loudspeaker icon. Earphones were handed out if necessary. All children filled out the questionnaire during a school lesson, with help from a research assistant or a teacher if needed. A paper version of the questionnaire was handed out if technical problems occurred.

### 2.6. Data Analysis

The SOPLAY observations were typed into a Microsoft Excel database immediately after each observation by the person who performed the observation. After data collection was completed, the database was transformed into a dataset and uploaded to the statistical program IBM SPSS Statistics 26 (IBM, New York, USA). Random quality controls were performed to assess whether any typing mistakes had occurred when data was typed into the database. Descriptive analyses were performed to examine the proportion of children observed in different PA levels and their smartphone usage during recess at baseline and follow-up. Pearson chi-square analyses was used to test for independence at a 0.05 significance level for school, grade, and gender between baseline and follow-up. Multinomial logistic regression analysis was performed to test whether PA intensity (sedentary, moderate, vigorous) changed between baseline and follow-up across all six schools, as well as to test for variation within schools (1–6), grades (4–7), and gender (male vs female). No significant change in weather and temperature was observed between baseline and follow-up, which is why they were not included in the regression model.

Questionnaire data were downloaded in a csv-format from the electronic survey program SurveyXact and imported into Stata version 16 to perform analysis. The dataset was initially cleaned for any obvious typing mistakes. Regarding the question “How frequently do you use a smartphone during recess on a weekly basis?”, a new yes/no category was created. This was done by combining the categories “1 day a week”, “2–3 days a week”, and “4–5 days a week” into a “Yes, I use a smartphone during recess on a weekly basis”, while the last category “never” was transformed into a “No, I never use a smartphone during recess”. Since the children were asked how frequently they did PA during the morning break and lunch break separately, a new variable was created using the mean across the two breaks to analyze the children’s general frequency of recess PA. The new variable containing the mean frequency of recess PA covered an ordinal scale of ten categories going from “1 = I am never active during recess” to “10 = I am always active during recess”. Descriptive analyses were performed to examine the self-reported mean frequency of weekly PA during recess at baseline and follow-up. An independent paired t-test was performed to test for significant changes in the weekly mean frequency of recess PA between baseline and follow-up for school, grade, and gender. Ordered logistic regression analysis was used to test whether weekly recess PA significantly changed between baseline and follow-up, and to test for variation within schools (1–6), grades (4–7), and gender (male vs female).

### 2.7. Ethics

An information letter was sent to all teachers and parents to inform them and the children about the study. A passive informed consent procedure was used, where the children were included unless the children or their parents withdrew consent. This procedure has been found to be ethically appropriate in low-risk research in the target group [23], and is in accordance with Danish regulations. The children could easily withdraw from the research part of the study without providing any explanation. The study was approved by the Research and Innovation Organization (RIO), University of Southern Denmark (10.790), and conducted in accordance with the World Medical Association Declaration of Helsinki.

## 3. Results

Out of 933 children from six different schools, a total of 814 children at baseline (90.3%) and 828 children at follow-up (91.6%) participated in the study and filled out the questionnaire. Furthermore, a total of 3437 children were observed at baseline and 3999 at follow-up, across all six participating schools (Table 2).

Both observational and self-reported data reached an equal distribution in gender at both baseline and follow-up. Grade 4 was the smallest participating group for both observational and self-reported data, whereas grades 5 and 6 were the largest. Primarily sedentary and moderate PA levels were observed at baseline and follow-up, and mean values of self-reported PA frequency were relatively high at both baseline and follow-up. Smartphone usage was different between observed and self-reported data; however, both data sources had a higher percentage of smartphone usage at baseline compared to follow-up (Table 2). At baseline, 82% girls, 67% boys, 74% fourth graders, 74% fifth graders, 71% sixth graders, and 78% seventh graders reported using a smartphone a minimum of one day per week during recess (data not shown in Table 2).

### 3.1. Physical Activity Intensity

The results of the multinomial logistic regression analysis are presented in Table 3. Both the estimated multinomial logistic regression coefficients (B), the standard errors (Std. Error), the significance level (Sig.), the odds ratio (Exp(B)), and the confidence intervals (95% confidence interval for Exp(B)) are presented.

The likelihood ratio tests showed statistical significance at <0.001 for all included factors (time, school, grade, and gender). The PA level “sedentary” was set as the reference category in the regression model, which is why only results for “moderate” and “vigorous” PA are presented in Table 3. Comparing baseline and follow-up (time), an odds ratio of 1.387 suggested that observed children were more likely to be active at a moderate level at follow-up, compared to being sedentary at follow-up. This was the opposite for vigorous PA level (odds ratio = 0.851). Some variation was observed between schools. Children at schools 2 and 3 were significantly less likely to be moderately active, as well as school 5, although not significant (*p*-value = 0.171). For vigorous PA, this was somewhat different, as only children at school 3 were less likely to be vigorously active (odds ratio = 0.637, *p*-value = 0.336), whereas the other schools were more likely to be vigorously active than the comparison school (school 6). Grade 5 children were most likely to be active on a moderate level (odds ratio = 1.610), whereas grade 4 children were most likely to be active on a vigorous level (odds ratio = 4.265). When comparing gender, boys were significantly more likely to be moderately (odds ratio = 1.836) and vigorously (odds ratio = 2.350) active than girls.

The results of the chi-square analysis comparing the percentage of change from baseline to follow-up for different subgroups are presented in Table 4.

All chi-square tests yielded a significance level <0.001, suggesting that the observed percentage of change in PA intensity from baseline to follow-up for each subgroup analysis was significant. Five schools were observed to decrease sedentary behavior, whereas one school (school 2) increased sedentary behavior (18.7%). In addition, five schools increased in moderate PA, whereas one school (school 1) slightly decreased moderate PA (−1.1%). Four schools increased in vigorous PA, whereas two schools (school 2; −80.2%, and 4; −44%) decreased vigorous PA. All grades significantly increased their moderate PA; however, only grade 4 slightly increased their vigorous PA (4.0%), whereas grades 5–7 decreased, of which grade 7 decreased substantially (−76.0%). Girls were observed being more sedentary at both baseline (51.0%) and follow-up (43.4%) than boys (33.0% and 31.5%, respectively), but the percentage decrease in sedentary behavior was much greater for girls (−15.1%) than for boys (−4.5%). Moderate PA significantly increased from baseline to follow-up for both boys (15.5%) and girls (18.7%), whereas vigorous PA decreased for both girls (−11.1%) and boys (−38.7%).

### 3.2. Frequency of Physical Activity

The results of the ordered logistic regression analysis are presented in Table 5. The estimated coefficients (B), the standard errors (Std. Error), the significance level (Sig.), the odds ratio (Exp(B)), and the confidence intervals (95% confidence interval for Exp(B)) are presented.

The likelihood ratio tests showed statistical significance at <0.001 for all included factors (time, school, grade, and gender). Comparing baseline and follow-up (time), an odds ratio of 1.370 suggested that children perceived themselves to be more frequently active at follow-up, compared to baseline (significant *p*-value = 0.002). Children at school 3 and school 6 were less frequently active than school 1, although only school 3 was significant (*p*-value = <0.001). Grade 4 children were most likely to report being frequently active compared to grade 5 (odds ratio = 0.844), grade 6 (odds ratio = 0.914), and grade 7 (odds ratio = 0.478). Grade 7 children were the only one being significantly less frequently active than grade 4 children (*p*-value = <0.001). When comparing gender, boys were significantly (*p*-value = <0.001) more likely to be frequently active than girls (odds ratio = 0.604).

The results of the *t*-test analysis comparing change from baseline to follow-up for different subgroups are presented in Table 4. The children reported rather high mean frequencies of recess PA across schools, gender, and grades, reporting from 6.7–9.2 at baseline and from 7.8–9.0 at follow-up. Four out of the six schools increased in their mean frequency of recess PA from baseline to follow-up. Two schools had a significant increase of 8.1% (school 2) and 13.9% (school 3), respectively. None of the schools had a significant decrease. All grades increased in mean frequency of recess PA, whereas only grade 6 children increased significantly (7.1%). Frequency of recess PA was also increased for both boys (3.5%) and girls (3.2%) from baseline to follow-up with only boys having a significant increase (*p*-value < 0.05).

## 4. Discussion

The aim of this study was to investigate if a ban on smartphone usage during recess changed children’s PA. Although vigorous PA decreased, the study overall revealed a significant increase in PA during recess—both in intensity and frequency—when a ban on smartphone usage was implemented for four weeks. Literature suggests that public health strategies, aiming at improving children’s health, should strive for increasing moderate PA rather than vigorous PA [24,25]. The increase in moderate PA was found across a broad range of schools strengthening the effect of the intervention. The findings are discussed in more detail in the following sections, including a discussion on a practical level to accommodate school boards’, health professionals’, and politicians’ request for this knowledge to inform future policies and actions [16,17].

### 4.1. Grade Differences

A difference in PA intensity was observed across grades. Grade 7 children were the least active on a moderate and vigorous level, whereas grade 5 children were the most active on a moderate level and grade 4 children were the most active on a vigorous level. From the self-reported data, we also found grade 4 children to be more frequently active than 5, 6, and 7 graders. Literature on recess PA and grade is not consistent. A review by Ridgers et al. (2012) found no association between grade and recess PA [13]. In contrast to the review, but more in line with our results, a large study containing 130 schools and 16,471 children from grades 4–10 found a difference in recess PA among the grades, with grade 6 being the most active during recess, following a decline from grade 7 [26]. As an explanation for grade differences in PA, a qualitative study concluded that children from grade 7 often felt lazier and more afraid of sweating compared with children from grade 5 and opted to stay sedentary during recess for these reasons [27]. Another explanation could be a more extensive usage of smartphones among grade 7 children. In a study on Brazilian children, a reduction in all day PA and an increase in all day screen use was reported by the children with increased age [28]. We also found a higher self-reported smartphone usage during recess among grade 7 children (78%) compared to grades 4–6 (71–74%), though the difference was not significant, and the use of smartphones was reported to be relatively high among all grades. When smartphones were banned during recess, all grades were observed to significantly increase their moderate PA which also indicates that usage of smartphones during recess influence PA among a broad age range of children. Previously, recess interventions have found to have the highest effect among the youngest children, because the initiatives employed often appealed more to the youngest children [29]. Finding a more equal effect among the grades in the current study suggests that an initiative such as a ban on smartphone usage during recess influences a broad age range of children.

### 4.2. Gender Differences

Our study revealed that boys were more active than girls—both in intensity and frequency. A gender difference has been consistently found in recess PA, with boys engaging in more PA than girls [13,30,31]. Therefore, attention in recent years has been on how to increase girls’ recess PA to reduce the gender difference in PA [32]. Most intervention studies have found a higher increase in recess PA among boys than girls and thus failed in reducing the gender difference [33,34,35]. The current study further revealed that girls (82%) reported to use smartphones more than boys (76%) during recess. A review also found girls to be prone to a higher smartphone addiction risk than boys [36]. To our knowledge, gender differences in smartphone usage during recess has not previously been investigated, and whether the girls’ lower recess PA is caused by their smartphone usage during recess is unknown. However, notably, we observed a much greater decrease in sedentary behavior and a slightly larger increase in moderate PA for girls than for boys when a ban on smartphone usage was implemented. It is also worth noting that particularly boys decreased in vigorous PA. They might have changed their PA behavior in follow-up because more girls participated in PA activities, resulting in more children in the same space. In relation, this could also have effected boys’ movement behavior through increased stress or lack of facilities in the schoolyard, which is a common mentioned barrier for recess PA [11,14]. Boys and girls reported an equal increase in the frequency of recess PA. Thus, the current study indicates that a ban on smartphone usage during recess increases PA (except vigorous PA) among both boys and girls, but that in particular girls decrease their sedentary behavior.

### 4.3. Practical Implications When Implementing a Ban on Smartphone Usage during Recess

The current study suggests that a ban on smartphone usage during recess would promote recess PA. A large Norwegian study, including 68 schools, also found interventions on the policy level (e.g., a written policy for PA and organized non-curricular PA) to be effective in promoting recess PA [37]. For a successful implementation of policies in schools, support from school staff is needed [38]. School policies are usually planned by individuals in the upper levels of the school, the municipality or the state, but are implemented by the teachers at the lower levels in the organization [39]. If individuals at the government, municipality, and school management levels do not listen to the teachers to understand the complexity of implementation, the policy may not be implemented as intended [40]. Even though it can be argued that a ban on smartphone usage during recess has a low complexity level, all individuals involved must be supportive and take responsibility for a ban on smartphone usage to be successfully implemented. In the current study, our analysis indicated that a relatively high percentage (32.4%) of children used their smartphone during follow-up due to implementation challenges. In our intervention, the teachers had a key role by being responsible for ensuring that all children handed in their smartphones, and for placing the smartphones in a locked cupboard. We informed the teachers but did not include them in the planning of the intervention, which might have caused a lack of support from some teachers. It is therefore suggested that implementing a ban on smartphone usage would benefit from being planned in close collaboration with both school management and teachers.

Five of six schools in our study had a policy obliging children to stay outdoors all recesses, half of each recess, or at lunch break due to the COVID-19 pandemic. The only school not having an outdoor policy during recess (school 3) was the school with the highest level of observed sedentary behavior and lowest level of self-reported mean frequency of recess PA at baseline. This is in line with a study finding that children are more sedentary indoors than outdoors during recess [41]. Interestingly, school 3 also had the highest increase in self-reported mean frequency of recess PA from baseline to follow-up, indicating that implementing a ban on smartphone usage in schools with no outdoor policy will be most effective. Nevertheless, since most schools decreased in sedentary behavior and increased in both observed and self-reported PA, our study showed that an outdoor recess policy did not eliminate the effect of the intervention. Thus, it suggests that implementing a ban on smartphone usage will be effective for schools no matter what, but that schools with no outdoor recess policy might get greater benefits from the intervention.

### 4.4. Strengths and Limitations

The strength of this study was the fairly large sample size and high participation rate (at baseline 90.3% and at follow-up 91.6%). Another strength was that we followed the same cohort of children over time. However, we did not collect individual identifiable information on the children and therefore we could not detect any changes that might have occurred over time on an individual level. Further, we did not include control schools because schools are entities with unique characteristics, capacities, and culture, which decrease the comparability between two samples of schools [42,43].

We used a questionnaire to complement the direct observations with self-reported data from the children. Using two methods complementing each other facilitated us in attaining a much richer form of data and greater credibility of results [44]. Still, both methods could have been improved by testing for inter-rater reliability amongst observations and test/re-test reliability for the questions asked in the questionnaire. Children within the target population are capable of accurately and reliably reporting their own health behavior [45]. However, the responses to the self-reported questions might have been driven by social desirability bias [46] that might have overestimated PA and underestimated the true quantity of smartphone usage [47,48], although the high percentage of self-reported smartphone usage at follow-up (32.4%) was assessed to be due to implementation challenges rather than participants not answering the survey question correctly, suggesting no social desirability bias.

While all likely areas for observation at each school were targeted, it is possible for PA and smartphone usage to have occurred in a space not observed by the research assistants [49], which can explain the difference between observed and self-reported data on smartphone usage. Nevertheless, comparisons can be made among different environments and within the same environment over time, which was the purpose of the current study. Other influencing factors not measured might also have had an impact on the assessed PA, for example, children’s individual health status and the school environment. Thus, the results should be interpreted with caution.

Finally, the participants were recruited from six schools diverse in type, size, geographical location, and parental socioeconomic status. This broad sample strengthened the possibility of generalization, although it is important to notice that the current study was conducted during the COVID-19 pandemic, having influenced the recess context. Further, seasonal variation in PA during recess must be considered when attempting to generalize the results. Data collection took place from August to October and children’s PA patterns during the other months of the year might yield different results [50]. A future intervention study (post COVID-19 pandemic) with a longer intervention period and using an objective measurement tool following the individual child (e.g., accelerometer) is warranted.

## 5. Conclusions

To our knowledge, this is the first intervention study investigating if a ban on smartphone usage during recess changed schoolchildren’s PA behavior. The paper filled an important gap in the literature for researchers, school boards, teachers, health professionals, and politicians on how schoolchildren’s PA during recess can be positively changed by policy. Findings from this novel intervention study revealed that as a whole, the children’s intensity and frequency of recess PA increased when implementing a ban on smartphone usage. Interestingly, the increase in PA was found among both schools having outdoor and indoor recess, among both boys and girls, and nearly equally among grades 4–7. This suggests that a policy initiative such as a ban on smartphone usage during recess would improve the everyday conditions for health among a broad range of schoolchildren. Future studies are needed to further investigate the association between recess PA and smartphone usage.

## Figures and Tables

**Table 1 ijerph-18-01907-t001:** Main characteristics of the six participating schools.

School No.	Type	Region	Area	No. on Roll	Parents’ Income Range *	Grade	Outdoor Recess Rule
1	Public	Central Denmark	Rural	<400	<Average	4–6	Yes
2	Public	Southern Denmark	Urban	>600	Average	5–7	Yes, grade 5–6 Grade 7 during lunch break
3	Private	Southern Denmark	Urban	400–600	>Average	5–7	No
4	Public	Southern Denmark	Urban	>600	>Average	4–7	Yes
5	Public	Northern Denmark	Rural	<400	<Average	4–6	Yes, half of each recess
6	Public	Northern Denmark	Urban	400–600	Average	4–6	Yes, half of each recess

* Income range of the parents having children attending the respective school. Published data from Statistics Denmark.

**Table 2 ijerph-18-01907-t002:** Descriptive statistics of baseline and follow-up measures for self-reported and observational data.

		Baseline		Follow-Up	
		Observational (Range%)	Self-Reported (Range%)	Observational (Range%)	Self-Reported (Range%)
N		3437 (303–815) ^a^	814 (55–282) ^a^	3999 (370–1030) ^a^	828 (55–296) ^a^
Gender	Boys	50.8% (44.3–55.3)	50.7% (38.2–55.9)	51.7% (45.4–57.0)	51.5% (36.4–57.5)
Grade	4	15.9% (0 *–43.6)	18.3% (0 *–40)	14.4% (0 *–38.5)	18.2% (0 *–34.6)
	5	29.6% (16.4–40.4)	30.2% (27.3–33.3)	28.7% (15.4–36.7)	30% (26.3–36.7)
	6	31.8% (24.3–38.2)	28.8% (19.2–43.0)	35% (17.5–50.7)	28.6% (18.7–40.0)
	7	22.7% (0 *–45.4)	22.7% (0 *–49.7)	22% (0 *–45.8)	23.2% (0 *–48.4)
Smartphone usage	Yes	18.0% (8.5–29.6)	74.1% (62.5–86.5)	0.3% (0.0–1.1)	32.4% (22.5–39.4)
PA intensity	Sedentary	41.9% (33.3–58.9)	-	37.2% (25.0–53.9)	-
	Moderate	43.1% (27.4–55.4)	-	52.1% (40.4–58.9)	-
	Vigorous	15.0% (6.1–28.8)	-	10.7% (5.7–20.3)	-
PA frequency	Mean ^b^	-	8.5 (6.7–9.2) ^c^	-	8.8 (7.8–9.0) ^c^

Note: % = percent. PA = physical activity. * Some schools did not include grade 4 or grade 7. ^a^. Shown as total numbers and not in percentages. ^b^. Frequency was measured using an ordinal scale going from 1 = “I am never active during recess” to 10 = “I am active during every recess”. ^c^. Shown as mean values and not in percentages.

**Table 3 ijerph-18-01907-t003:** Multinomial logistic regression analysis for physical activity intensity.

Physical Activity Intensity ^a^		B	Std. Error	Sig.	Exp(B) ^c^	95% Confidence Interval for Exp(B)	
						Lower Bound	Upper Bound
Moderate	Intercept	−0.392	0.098	0.000			
	Time						
	Baseline	0 ^b^					
	Follow-up	0.327	0.052	0.000	1.387	1.254	1.536
	School						
	1	0.244	0.097	0.012	1.276	1.056	1.542
	2	−0.693	0.088	0.000	0.500	0.420	0.595
	3	−0.514	0.087	0.000	0.598	0.505	0.709
	4	0.273	0.083	0.001	1.314	1.117	1.546
	5	−0.139	0.102	0.171	0.870	0.713	1.062
	6	0 ^b^					
	Grade						
	4	0.219	0.103	0.033	1.245	1.018	1.522
	5	0.476	0.078	0.000	1.610	1.381	1.877
	6	0.235	0.077	0.002	1.265	1.089	1.470
	7	0 ^b^					
	Gender						
	Boys	0.608	0.052	0.000	1.836	1.659	2.033
	Girls	0 ^b^					
Vigorous	Intercept	−2.686	0.157	0.000			
	Time Baseline	0 ^b^					
	Follow-up	−0.161	0.077	0.037	0.851	0.732	0.990
	School						
	1	0.749	0.141	0.000	2.115	1.605	2.786
	2	0.726	0.129	0.000	2.067	1.606	2.660
	3	−0.148	0.154	0.336	0.862	0.637	1.167
	4	0.943	0.126	0.000	2.568	2.008	3.285
	5	0.385	0.149	0.010	1.470	1.098	1.968
	6	0 ^b^					
	Grade						
	4	1.450	0.148	0.000	4.265	3.190	5.702
	5	0.849	0.121	0.000	2.337	1.845	2.960
	6	0.694	0.120	0.000	2.001	1.580	2.534
	7	0 ^b^					
	Gender						
	Boys	0.854	0.079	0.000	2.350	2.014	2.742
	Girls	0 ^b^					

^a^. The reference category is “sedentary”; ^b^. This parameter is set to zero because it is redundant; ^c^. Exp(B) are the odds ratios for the predictors.

**Table 4 ijerph-18-01907-t004:** Proportion of children observed in different physical activity intensities, and mean frequency of physical activity from self-reported questionnaire data, baseline–follow-up.

	Physical Activity Intensity				Physical Activity Frequency ^a^
	Sedentary (Percentage Change)	Moderate (Percentage Change)	Vigorous (Percentage Change)	Total (Count)	Mean (Percentage Change)
School					
1	33.3–25.0 (−24.9)	55.4–54.8 (−1.1)	11.3–20.3 (44.3)	419–416	8.6–8.9 (3.6)
2	43.8–53.9 (18.7)	27.4–40.4 (32.2)	28.8–5.7 (−80.2)	681–705	8.2–8.9 (8.1) ^c^
3	58.9–45.7 (−22.4)	35.0–46.7 (25.0)	6.1–7.7 (20.8)	511–698	6.7–7.8 (13.9) ^c^
4	34.9–31.2 (−10.6)	47.3–58.9 (19.7)	17.8–9.9 (−44.4)	815–849	9.1–9.0 (−0.6)
5	45.0–31.9 (−29.1)	40.7–51.8 (21.4)	14.3–16.3 (12.3)	303–370	9.2–8.8 (−4.5)
6	40.1–32.1 (−19.9)	51.4–56.9 (9.7)	8.5–11.0 (22.7)	811–1030	8.6–8.8 (2.0)
Grade					
4	38.3–31.0 (−19.1)	42.7–49.2 (13.2)	19.0–19.8 (4.0)	548–575	8.9–9.0 (1.0)
5	37.6–32.9 (−12.5)	47.9–55.1 (13.1)	14.4–12.0 (−16.7)	1018–1146	8.7–8.9 (2.0)
6	45.1–33.9 (−24.8)	41.5–55.9 (25.8)	13.5–10.3 (−23.7)	1092–1400	8.3–9.0 (7.1) ^c^
7	45.4–52.4 (13.4)	39.5–44.0 (10.2)	15.0–3.6 (−76.0)	779–878	8.0–8.2 (2.4)
Gender					
Boys	33.0–31.5 (−4.5)	47.9–56.7 (15.5)	19.1–11.7 (−38.7)	1746–2067	8.6–8.9 (3.5) ^b^
Girls	51.0–43.4 (−15.1)	38.2–47.0 (18.7)	10.8–9.6 (−11.1)	1691–1932	8.3–8.6 (3.2)

Note: Chi-square tests for all observed data: *p*-value = <0.001; ^a^. Frequency was measured using an ordinal scale going from 1 = “I am never active during recess” to 10 = “I am active during every recess”; ^b^. Paired *t*-test for self-reported data: *p*-value < 0.05; ^c^. Paired t-test for self-reported data: *p*-value < 0.01.

**Table 5 ijerph-18-01907-t005:** Ordered logistic regression analysis for physical activity frequency.

Physical Activity Frequency ^a^		B	Std. Error	Sig.	Exp(B) ^c^	95% Confidence Interval for Exp(B)	
						Lower Bound	Upper Bound
	Time						
	Baseline	0 ^b^					
	Follow-up	0.315	0.101	0.002	1.370	0.117	0.514
	School						
	1	0 ^b^					
	2	0.210	0.215	0.329	1.233	−0.211	0.631
	3	−0.939	0.216	0.000	0.391	−1.361	−0.517
	4	0.650	0.195	0.001	1.915	0.267	1.033
	5	0.468	0.267	0.079	1.597	−0.055	0.991
	6	−0.054	0.204	0.791	0.947	−0.453	0.345
	Grade						
	4	0 ^b^					
	5	−0.169	0.163	0.298	0.844	−0.488	0.150
	6	−0.088	0.162	0.584	0.915	−0.405	0.228
	7	−0.737	0.184	0.000	0.478	−1.097	−0.378
	Gender						
	Boys	0 ^b^					
	Girls	−0.503	0.103	0.000	0.604	−0.704	−0.302

^a^. Frequency was measured using an ordinal scale going from 1 = “I am never active during recess” to 10 = “I am active during every recess”; ^b^. This parameter is set to zero because it is redundant; ^c^. Exp(B) are the odds ratios for the predictors.

## Data Availability

The data presented in this study are available on request from the corresponding author.

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
