# Peer review of "A Ban on Smartphone Usage during Recess Increased Children’s Physical Activity"

_ijerph, 2021, doi:10.3390/ijerph18041907_

Round 1

Reviewer 1 Report

There is considerable discussion about smartphones and similar technology contributing to sedentary behaviours in children and adolescence.  Recess in schools is valued as a time for a range of interactions, including physical activity.  The present study provides much needed evidence and a foundation for future research investigating the impact of smartphones and tablets on physical activity.

Table 1: Is parent income range correct or is it the income range for the area where the school is located?  I’m not sure how schools are accessed in Denmark but I wondered if it is possible that parent income would not always match the range for the school area (if that’s where the data comes from)

Line 26: ‘During the last years’ is not very clear.  This could be reworded.

Lines 134-135 it is mentioned that all observations were conducted by a trained research assistant. It seems like a lot of observations for one research assistant.  Were there multiple research assistants? It would be helpful to provide more details, particularly how the researchers ensured the observations were reliable.  Did any interrater reliability checks take place?

More detail on the questionnaire would be helpful e.g. was it a Likert scale, open questions etc?  There are some indications of questions later in the manuscript but it would be helpful to have a fuller description when it is initially introduced in the paragraph beginning Line 136. Were there any attempts to check reliability or Cronbach alphas? 

The discussion could include considerations of why vigorous PA reduced.  This is an interesting finding as it seems that children may have been trying to compress their PA into a shorter period of time when they had access to their phones.  I wondered if the SOPLAY observations might provide indications in differences in activities.  It might also be worth looking at studies such as “Skrede, T., Stavnsbo, M., Aadland, E., Aadland, K. N., Anderssen, S. A., Resaland, G. K., & Ekelund, U. (2017). Moderate-to-vigorous physical activity, but not sedentary time, predicts changes in cardiometabolic risk factors in 10-y-old children: The Active Smarter Kids Study. The American Journal of Clinical Nutrition, 105(6), 1391-1398.” that suggest moderate may be a more important focus than vigorous PA.

Table 2: Should the commas for PA frequency be decimal points?

Line 46: had = have

Line 105: study = the study

Line 140: lasted = required

Line 280: whereof = whereas

Line 301: 16.471 = 16,471

Author Response

Thank you for reviewing our manuscript. Our point-by-point response to your comments are below.

---

There is considerable discussion about smartphones and similar technology contributing to sedentary behaviours in children and adolescence.  Recess in schools is valued as a time for a range of interactions, including physical activity.  The present study provides much needed evidence and a foundation for future research investigating the impact of smartphones and tablets on physical activity.

Table 1: Is parent income range correct or is it the income range for the area where the school is located?  I’m not sure how schools are accessed in Denmark, but I wondered if it is possible that parent income would not always match the range for the school area (if that’s where the data comes from)

--> It refers to the income of the parents having children attending the specific school and not the income range for the area where the school is located. We have tried to clarify this in a note, lines 82-83.

Line 26: ‘During the last years’ is not very clear.  This could be reworded.

--> You are right. We have deleted that part of the sentence

Lines 134-135 it is mentioned that all observations were conducted by a trained research assistant. It seems like a lot of observations for one research assistant.  Were there multiple research assistants? It would be helpful to provide more details, particularly how the researchers ensured the observations were reliable. Did any interrater reliability checks take place?

--> There were multiple research assistants and we trained all research assistants on a workshop to ensure alignment and reliability, however we did not do any interrater reliability checks. We have clarified these points in the manuscript lines 135-137 and lines 396-398.

More detail on the questionnaire would be helpful e.g. was it a Likert scale, open questions etc?  There are some indications of questions later in the manuscript, but it would be helpful to have a fuller description when it is initially introduced in the paragraph beginning Line 136. Were there any attempts to check reliability or Cronbach alphas? 

--> We have included a fuller description in the introduction to the questionnaire, lines 140-149. We did not check reliability or Cronbach alphas. We have put in a comment on that in the strength and limitation section, lines 396-398.

The discussion could include considerations of why vigorous PA reduced. This is an interesting finding as it seems that children may have been trying to compress their PA into a shorter period of time when they had access to their phones.  I wondered if the SOPLAY observations might provide indications in differences in activities.  It might also be worth looking at studies such as “Skrede, T., Stavnsbo, M., Aadland, E., Aadland, K. N., Anderssen, S. A., Resaland, G. K., & Ekelund, U. (2017). Moderate-to-vigorous physical activity, but not sedentary time, predicts changes in cardiometabolic risk factors in 10-y-old children: The Active Smarter Kids Study. The American Journal of Clinical Nutrition, 105(6), 1391-1398.” that suggest moderate may be a more important focus than vigorous PA.

--> You are right that it is interesting that VPA decreased. Unfortunately, our observations cannot explain the reason, yet we have included a consideration like the one you mention, lines 343-348, but it will only be a conjecture - not supported by our data. We have also included your suggested paper, lines 297-299.

Table 2: Should the commas for PA frequency be decimal points?

--> Yes, corrected

Line 46: had = have --> corrected

Line 105: study = the study --> corrected

Line 140: lasted = required --> corrected

Line 280: whereof = whereas --> corrected

Line 301: 16.471 = 16,471 --> corrected

Reviewer 2 Report

  1. This paper was to investigate if the ban on smartphone usage during recess changed children's physical activity. However, of course, if students cannot use their smartphones, they do not stay still and do physical activities, and this can be given meaning only when some evidence that this improves their health status. Just this content alone seems to be a bit insufficient for the paper's contribution.

  2. In Table 3, the caption should represent multinomial regression analysis as multinominal logistic regression analysis. Incorrect terms in the caption of Table 3 or in the content of the body need to be corrected.

  3. Using Table 4, a chi-square test was conducted to see if there was a difference between each grade level in the physical activity of the students. It is necessary to express the result in detail. Insert the right column of the table 4, specify the chi-square test statistic and p-value, and describe the result.

Author Response

Thank you for reviewing our manuscript. Our point-by-point response to your comments are below.

  1. This paper was to investigate if the ban on smartphone usage during recess changed children's physical activity. However, of course, if students cannot use their smartphones, they do not stay still and do physical activities, and this can be given meaning only when some evidence that this improves their health status. Just this content alone seems to be a bit insufficient for the paper's contribution.

--> Of course, it could be a hypothesis that if students cannot use their smartphones, they do not stay still and do physical activities. However, as far as we know, our study is the first study to test this hypothesis. Further, from our study and previous experiences in this research field, we believe that it cannot be seen as simple as you dispose it. In the study we still observe sedentary behavior during follow-up although we did not observe smartphone usage. Some still did sedentary activities such as reading, painting, playing cards etc. Interestingly, we found and discussed differences between gender, grades and schools which is important knowledge that needs to be further investigated.

  1. In Table 3, the caption should represent multinomial regression analysis as multinominal logistic regression analysis. Incorrect terms in the caption of Table 3 or in the content of the body need to be corrected.

--> We have corrected the term to “multinomial logistic regression analysis” throughout the manuscript. We have also changed the term “ordered logistic regression analysis” by including “logistic”.

  1. Using Table 4, a chi-square test was conducted to see if there was a difference between each grade level in the physical activity of the students. It is necessary to express the result in detail. Insert the right column of the table 4, specify the chi-square test statistic and p-value, and describe the result.

--> We are not sure if we understood your comment correctly, but the chi-square test was not conducted to see if there was a difference between each grade level in the physical activity of the students. For this analysis we used the multinomial logistic regression analysis presented in Table 3. As written in the data analysis section lines 164-165 the chi-square testwas used to test for independence at a 0.05 significance level between baseline and follow-up. As written on line 249, all chi-square tests achieved a p value < 0.001. Which is why we decided not to specify this in table 4. We are not sure which other column you want us to insert in table 4.  

Round 2

Reviewer 2 Report

It seems to have confirmed the natural result through an experiment. Taking advantage of government policy with this seems insufficient. This paper requires a variety of analyzes, including more attributes.
"When children are banned from using their smartphones during recess,
I think that their health, weather, temperature, and the surrounding environment can also affect their physical activity.
So, if you use that many attributes when doing a survey, I think you can do more diverse analysis. 
The attributes used in the authors' research are also important, but using more attributes will yield better results.
This paper also has a lot of meaning."

Author Response

It seems to have confirmed the natural result through an experiment. Taking advantage of government policy with this seems insufficient. This paper requires a variety of analyzes, including more attributes.
"When children are banned from using their smartphones during recess,I think that their health, weather, temperature, and the surrounding environment can also affect their physical activity.So, if you use that many attributes when doing a survey, I think you can do more diverse analysis. The attributes used in the authors' research are also important but using more attributes will yield better results.This paper also has a lot of meaning.

--> You are right that other factors, as you mention, can influence children’s recess PA. We included data on weather and temperature in our observations. However, no significant change in weather and temperature was observed between baseline and follow-up, which is why they were not included in the regression model, mentioned in the manuscript, lines 127-129 + 171-173. Unfortunately, it is not possibleto add further variables to our research and we will add that as a limitation in the study, lines 447-449. Importantly, we believe, as also mentioned in the manuscript, that our findings need to be further investigated in future studies.